# Maternal Inflammatory Biomarkers during Pregnancy and Early Life Neurodevelopment in Offspring: Results from the VDAART Study

**DOI:** 10.3390/ijms232315249

**Published:** 2022-12-03

**Authors:** Rachel S. Kelly, Kathleen Lee-Sarwar, Yih-Chieh Chen, Nancy Laranjo, Raina Fichorova, Su H. Chu, Nicole Prince, Jessica Lasky-Su, Scott T. Weiss, Augusto A. Litonjua

**Affiliations:** 1Channing Division of Network Medicine, Brigham and Women’s Hospital and Harvard Medical School, Boston, MA 02115, USA; 2Division of Allergy and Clinical Immunology, Brigham and Women’s Hospital and Harvard Medical School, Boston, MA 02467, USA; 3Laboratory of Genital Tract Biology, Department of Obstetrics, Gynecology and Reproductive Biology, Brigham and Women’s Hospital and Harvard Medical School, Boston, MA 02115, USA; 4Division of Pediatric Pulmonary Medicine, Golisano Children’s Hospital at Strong, University of Rochester Medical Center, Rochester, NY 14642, USA

**Keywords:** IL-8, CRP, ages and stages questionnaire, pregnancy, neurodevelopment, inflammation

## Abstract

Maternal infection and stress during the prenatal period have been associated with adverse neurodevelopmental outcomes in offspring, suggesting that biomarkers of increased inflammation in the mothers may associate with poorer developmental outcomes. In 491 mother–child pairs from the Vitamin D Antenatal Asthma Reduction Trial (VDAART), we investigated the association between maternal levels of two inflammatory biomarkers; interleukin-8 (IL-8) and C-Reactive Protein (CRP) during early (10–18 wks) and late (32–38 wks) pregnancy with offspring scores in the five domains of the Ages and Stages Questionnaire, a validated screening tool for assessing early life development. We identified a robust association between early pregnancy IL-8 levels and decreased fine-motor (β: −0.919, 95%CI: −1.425, −0.414, *p* = 3.9 × 10^−4^) and problem-solving skills at age two (β: −1.221, 95%CI: −1.904, −0.414, *p* = 4.9 × 10^−4^). Associations between IL-8 with other domains of development and those for CRP did not survive correction for multiple testing. Similarly, while there was some evidence that the detrimental effects of early pregnancy IL-8 were strongest in boys and in those who were not breastfed, these interactions were not robust to correction for multiple testing. However, further research is required to determine if other maternal inflammatory biomarkers associate with offspring neurodevelopment and work should continue to focus on the management of factors leading to increases in IL-8 levels in pregnant women.

## 1. Introduction

Maternal health and environment during pregnancy play a crucial role in the early life neurodevelopment of offspring, and systemic inflammation in the mothers has been shown to be among the most important risk factors [1]. Such inflammation may be the result of bacterial or viral infections and environmental exposures [2,3], but can also be due to certain medical conditions of pregnancy including preeclampsia and elevated body mass index, or to other prenatal stressors; both common daily stressors and more extreme levels of stress during pregnancy have been associated with adverse neurodevelopment in the children [4,5,6].

Inflammation is typically measured by quantifying circulating cytokines and chemokines and other proinflammatory molecules, such as C-reactive protein (CRP) in the blood [7]. Cytokines, a family of soluble polypeptides, including the chemokine subclass, contribute to the initiation, maintenance, and regulation of inflammatory immune responses to infection and insult [6,7]. C Reactive Protein (CRP), an acute phase protein made by the liver, shows increased levels in response to inflammation following the secretion of the proinflammatory cytokine IL-6 by macrophages and T cells [8].

Several studies have linked prenatal measures of inflammatory biomarkers with adverse neurodevelopmental outcomes in offspring including cognitive ability [9], neurodevelopmental delay [1], psychomotor development [10], autism [11,12,13,14,15], neurologic abnormalities [4], depression [16], psychosis [17] and schizophrenia [6]. Although directions of effect have not always been consistent across these studies, there is a strong biological rationale for the involvement of the dysregulation of maternal immune function and maternal inflammatory biomarkers in childhood neurodevelopment [4]. Cytokines have been shown to influence several diverse aspects of typical neurodevelopment, including proliferation and differentiation of neural and glial cells [18], which may explain observed differences in brain morphology and behavior of offspring of mothers with high levels of these biomarkers across the prenatal period [13]. Similarly, CRP has been observed to increase the permeability of the blood–brain barrier through its binding to the Fcγ receptor, resulting in the activation of microglia in the brain [19,20], and thereby influencing neuronal plasticity.

In this exploratory study, we had available measures for two inflammatory biomarkers in the Vitamin D Antenatal Asthma Reduction Trail (VDAART) mother child-cohort [21]; interleukin-8 (IL-8) and CRP, during early and late pregnancy in a mother–child cohort. We sought to determine if maternal levels of these biomarkers associated with early life child neurodevelopment as assessed by the Ages and Stages questionnaire (ASQ) [22] at ages one, two and three, with the hypothesis that higher levels of inflammation in the mothers, as measured by circulating IL-8 and CRP, associate with decreased performance across the five developmental domains of the ASQ in their children. We further explored whether these associations were influenced by offspring sex, breastfeeding, vitamin D supplementation or preterm birth based on reported roles for these variables in neurodevelopment [23,24,25,26,27], and whether the observed association between maternal BMI and child neurodevelopment [28,29] is mediated by the measured inflammatory biomarkers. This represents the first study to explore the potential influence of prenatal CRP and IL-8 on developmental outcomes as evaluated across the full breadth of the ASQ, which can be used as a screening tool to identify children in need of intervention. It aims to confirm previous reports that maternal inflammation across pregnancy, measured via levels of inflammatory biomarkers, can impact early childhood development.

## 2. Results

### 2.1. Study Participants

A total of 491 mother–child pairs with CRP and IL-8 measured during pregnancy and completed Age and Stages Questionnaires (ASQ) at age 1, 2 or 3 years were included in these analyses. The characteristics of the study sample are shown in Table 1. The greatest percentage of mothers were non-Hispanic Black (38.1%); the majority were not college graduates (62.5%) and were married or cohabiting (74.3%). Roughly half (48.3%) were assigned to the intervention arm in the original vitamin D supplementation trial [21]. A majority of mothers had been previously pregnant at the time of recruitment (64.8%) and just over half (58.3%) had a living child. The mean gestational age at delivery was 39.2 weeks (SD: 1.59 weeks); and 49.3% of the offspring were female. The characteristics of this subpopulation did not significantly differ from the total parent population (n = 881 mothers and 817 offspring) with the exception of gestational age which was significantly older than in the full population (*p* = 0.007) (Appendix A).

The mean log-transformed CRP levels in the mothers were 2.09 mg/L (SD 1.122 mg/L) and 2.02 mg/L (SD 1.042 mg/L) at enrollment (early pregnancy) and the third trimester (late pregnancy) respectively. The mean log transformed IL-8 levels were 1.38 pg/mL (SD 1.36 pg/mL) and 1.25 pg/mL (1.10 pg/mL) at the same timepoints. CRP levels measured during early and late pregnancy were highly correlated with each other (Pearson rho = 0.677, *p* < 0.001), while there was a weak but significant correlation between the two timepoints for IL-8 (rho = 0.160, *p* < 0.001). CRP and IL-8 were not correlated with each other during early pregnancy, but we did observe a significant weak correlation between the two markers during late pregnancy (rho = 0.124, *p* = 0.006) (Appendix A).

Of the 491 offspring, 399, 440 and 463 had completed ASQs at ages, one, two, and three respectively; 360 had completed questionnaires at all three years (Appendix A). There was moderate to poor correlation between the different domains of the ASQ, that tended to be stronger within years than between them, suggesting children’s level of development may differ across the first three years of life (Appendix A). Out of a total possible score of 60, the mean score for every domain was >44, corresponding to the majority of children (>79% for every domain at each year) being classified as on schedule for normal development (Appendix A). For any given age, the number of children in the lower two categories (requiring monitoring; requiring follow-up) for every domain was very small; two children at age one, three children at age two and five children at age three. Overall, these findings suggest little evidence of widespread developmental delays in this population.

### 2.2. Maternal Inflammatory Biomarkers and ASQ Scores

The associations between levels of IL-8 and or CRP with the five domains of the ASQ at ages one, two and three, after adjustment for study site, original treatment assignment, maternal race, maternal marital status, maternal education and gestational age at delivery are shown in Figure 1 and Appendix A. Nominally significant associations (*p* < 0.05) were observed between higher maternal levels of IL-8 during early pregnancy with lower gross motor (β: −0.902, 95%CI: −1.697, −0.107, *p* = 0.026) and fine motor (β: −0.880, 95%CI: −1.500, −0.259, *p* = 0.006) skills score at age one. This persisted for fine motor skills score at age two (β: −0.919, 95%CI: −1.425, −0.414, *p* = 3.9 × 10^−4^). Age two scores for personal social skills (β: −0.844, 95%CI: −1.452, −0.237, *p* = 0.007) and problem solving (β: −1.221, 95%CI: −1.904, −0.414, *p* = 4.9 × 10^−4^) were also inversely associated with maternal IL-8 levels during the early pregnancy. Late pregnancy IL-8 was not associated with any ASQ domains at any age.

Higher maternal levels of CRP during early pregnancy were associated with a nominally significantly lower problem-solving score at age one (β: −1.028, 95%CI: −2.022, −0.035, *p* = 0.043). Higher levels of CRP during both early and late pregnancy were associated with lower communication skills scores at age three; β: −0.64, 95%CI: −1.265, −0.015, *p* = 0.045 and β: −0.725, 95%CI: −1.401, −0.048, *p* = 0.036 for the early and late pregnancy, respectively. However, after applying the Benjamini–Hochberg FDR to account for multiple comparisons, only the associations between early IL-8 with fine motor skills score at age two and problem-solving score at age two retained significance (FDR *p*-value = 0.015 and 0.015, respectively).

We also investigated the mothers of the seven children who performed poorly across all domains in at least one year. However, there was no systematic evidence that these mothers had higher levels of IL-8 nor CRP at either visit than the rest of the population (Appendix A)**.**

### 2.3. Change in Biomarker Levels across Pregnancy and ASQ Scores

We sought to determine whether a change in these inflammatory biomarkers across early to late pregnancy was associated with ASQ scores. In 271 (55.2%) mothers, IL-8 levels were higher in early than in late pregnancy, while in 220 (44.8%) IL-8 levels were lower in early as compared to late pregnancy. Similarly, for CRP levels 268 (54.6%) mothers displayed a decrease across pregnancy, while 223 (45.4%) showed an increase. There was no correlation between a mother’s change in IL-8 and her change in CRP across pregnancy (*r* = −0.08, *p* = 0.072, Appendix A).

We observed that an increase in IL-8 levels from early to late pregnancy was nominally associated with a higher fine motor skills score at age one (β: 0.547, 95%CI: 0.022, 1.073, *p* = 0.041) and age two (β: 0.675, 95%CI: 0.252, 1.099, *p* = 0.002 and an increase in problem solving skills at age 2 (β: 0.802, 95%CI: 0.229, 1.375, *p* = 0.006). However, when the change in level was adjusted for early pregnancy level the associations lost their significance. There were no significant associations between a change in CRP levels across pregnancy and ASQ skills scores.

### 2.4. Interactions between Maternal Inflammatory Biomarkers with Offspring Sex, Breastfeeding and Vitamin D Supplementation

There is a well-classified disparity in the prevalence of neurodevelopment disorders between girls and boys [24]. Therefore, we reran all models including a child sex–inflammatory biomarker interaction term. Nominally significant interactions were observed between child sex and maternal IL-8 levels during early pregnancy and age three personal-social score (Interaction β = 1.161; 95% CI: −2.124, −0.198, *p*-interaction = 0.019) and age three problem solving score (Interaction β = 1.395; 95% CI: −2.677, −0.113, *p*-interaction = 0.033) (Appendix A, Figure 2). In both cases, the inverse association between IL-8 and ASQ score was more pronounced in boys than girls. However, neither were robust to FDR adjustment and overall, there was little evidence for a sex-interaction effect.

Similarly, we investigated a potential interaction with exclusive breastfeeding up to age four months, based on previous reports of a protective effect of breast feeding on the ASQ scores of preterm children [30]. Information on breastfeeding status was available for 457 (93%) mother–child pairs. Of these, 157 (34.4%) mothers reported exclusive breastfeeding for the first four months of their offspring’s life. The only observed interaction was for personal social score at age two; Interaction β = 1.297; 95% CI: 0.067, 2.527, *p*-interaction = 0.039) which was not robust to correction for multiple testing (Appendix A, Figure 3).

As this study was originally designed as a clinical trial supplementing pregnant women with vitamin D or placebo [21], we further sought to determine if supplementation interacted with biomarker levels in ASQ scores (Appendix A). Only one significant interaction was observed (Figure 4) which suggested that the effects of CRP on age three personal social skills may be more detrimental in those whose mothers received the placebo than those who received vitamin D. However, again this was not robust to correction for multiple testing.

### 2.5. Sensitivity Analyses

Given the known association between maternal inflammatory biomarkers and preterm birth [31], we ran a sensitivity analysis excluding 34 mother–child pairs with a gestational age <37 weeks (Appendix A). The results were attenuated somewhat in this restricted dataset; however, all directions of effect were consistent and the associations between IL-8 with age two fine motor and problem-solving skills retained nominal significance.

Similar findings were observed when we additionally adjusted for pre-pregnancy BMI. Pre-pregnancy was the only measure of BMI available and was missing for 65 VDAART mothers. Associations were again attenuated and the associations between IL-8 with age two fine motor and problem-solving skills only nominally significant (Appendix A). We then investigated whether pre-pregnancy BMI was associated with ASQ score. Only age one problem solving skills was nominally association with prepregnancy BMI (β = −0.201; 95% CI: −0.363, −0.038, *p* = 0.016, (Appendix A) and, mediation analyses determined that neither CRP or IL-8 were mediators of this observed relationship (Appendix A).

## 3. Discussion

The prenatal environment is critical for early life development, which itself is an important predictor of later life function and health [32]. As such, the identification of prenatal factors that influence development in offspring is vital to understand the mechanisms behind these adverse outcomes and to develop strategies to prevent them.

In this study of nearly 500 mother–child pairs from the VDAART study, we measured two key inflammatory biomarkers available in our cohort, CRP and IL-8, at two timepoints across pregnancy, to determine if levels associated with metrics of development in offspring at ages one, two and three years, as measured by the ASQ. Overall, we report an association between IL-8 levels in early pregnancy and offspring fine motor and problem-solving skills at age two, as assessed by the ASQ, that was robust to correction for multiple testing and adjustment for potential confounders. However, we did not find strong evidence for an association of prenatal IL-8 with personal-social, communication or gross motor skills or for prenatal CRP with any domains of the ASQ.

Higher levels of IL-8 in early pregnancy (10–18 weeks) were associated with significantly lower fine motor skills and problem-solving skills at age two. There were no significant associations with late pregnancy (32–38 weeks) levels, and while an increase in IL-8 across pregnancy appeared to be associated with better fine motor and problem-solving skills, this disappeared when accounting for early pregnancy levels. This suggests that if IL-8 is influencing facets of early life development, the early pregnancy timepoint may be the most pivotal. This is in agreement with studies of pregnant animals reporting that the adverse developmental effects in offspring are most pronounced when stress, which is known to induce increased production of IL-8 [33], is applied early in gestation [5].

IL-8, an 8 kDa protein, is a proinflammatory chemokine, a subclass of the cytokine superfamily which contribute to the initiation, maintenance, and regulation of inflammatory immune response to stressors or infection [6]. It is also among the primary coactivators of hypothalamic–pituitary–adrenal (HPA) axis function and an indicator of stress–immune pathway activation [4]. During bacterial and viral infection, IL-8 is produced by macrophages, epithelial cells, airway smooth muscle cells and endothelial cells and functions primarily to induce chemotaxis in target cells causing them to migrate toward the site of infection; to stimulate phagocytosis and to promote angiogenesis [6]. As such, IL-8 levels in peripheral circulation provide a reliable indicator of inflammation and levels have been shown to increase in response to bacterial and viral infections, as well as oxidative stress, maternal stress and several conditions of pregnancy including preeclampsia, obesity, anemia and gestational diabetes [4,6,34]. We have previously shown within this cohort that IL-8 levels during pregnancy are associated with a history of premature births, lower maternal education, an unhealthy maternal diet and smoke exposure [31]. However, importantly, IL-8 is also an angiogenic factor with critical roles during healthy pregnancy, including vascular remodeling, blastocyst implantation, placental development, permeability of the fetal–maternal unit and labor onset [6].

Several conditions known to induce IL-8 production, such as socioeconomic adversity, maternal obesity, preeclampsia and maternal infections during pregnancy, have themselves been linked to adverse neurodevelopment among offspring [4,6,29,35], offering this proinflammatory chemokine as a potential explanatory mechanism. Chemokines can pass through the placenta and the blood–brain barrier and act to regulate the communication between neurons and microglia, the immune cells of the central nervous system [13]. The receptor for IL-8, *CXCR2* (C-X_C Motif Chemokine Receptor 2), is present in the human fetal brain and plays a key role in role in synaptogenesis; therefore, IL-8 has been postulated to direct the migration and differentiation of neural stem/progenitor cells which are crucial for early neural development [4]. High levels of IL-8 can lead to systemic inflammation which may disrupt the normal trafficking of neuronal processes, the formation of appropriate synaptic contacts and neural plasticity [4,6,13,36]. Accordingly, several studies have linked increased maternal inflammatory cytokines to increased risk of serious conditions including cerebral palsy, autism, schizophrenia, intellectual disability, depression and psychosis among offspring [1,4,6,9,10,11,12,13,14,15,16,17]. However, to our knowledge this is the first study to specifically link prenatal IL-8 to fine motor and problem-solving skills in early childhood.

The production of CRP, a non-specific biomarker of systemic inflammation, is also increased with a variety of stressors, including adverse pregnancy outcomes [37]. Within this population, we have previously observed associations between maternal CRP levels with lower maternal education, higher gravidity, uncontrolled asthma, unhealthy diet, prior premature births and risk of preterm birth [31]. A recent study within the VIVA cohort demonstrated that the observed association between maternal obesity with offspring visual motor abilities and fine motor skills was mediated by maternal CRP [38]. While work from the PREDO cohort considering a composite neurodevelopmental outcome, which included metrics from the ASQ, reported that higher levels of CRP across all of pregnancy associated with delays in the children [1]. However, in these analyses we did not observe any consistent associations between maternal CRP during early or late pregnancy and offspring personal-social, problem solving, fine-motor, gross-motor or communication skills in early life.

We considered several potential confounders and effect modifiers in these analyses. The same ASQ questionnaire is applied to both sexes and there is some disagreement as to whether there are differences in performance, with several studies reporting that girls tend to outperform boys overall [39]. In this study, we observed no strong evidence for an interaction between sex and inflammatory biomarkers on ASQ score. For those sex interactions that did meet statistical significance, early pregnancy IL8 with age three personal-social and problem-solving skills, the negative influence of IL-8 was greater in male than female offspring. Despite the reported protective effect of breast feeding on development skills [30,31], we did not observe evidence of this phenomena with the exception of age two personal-social skills, in which breastfeeding appeared to “rescue” the adverse effects of IL-8 exposure early in gestation. Similarly, we present weak evidence that vitamin D supplementation during pregnancy may be protective. However, it should be noted that than none of these interaction associations would be robust to correction for multiple testing and we were likely unpowered to detect interactions.

Maternal inflammatory biomarkers have been shown to be associated with preterm birth (gestational age at delivery <37 weeks) including within this population, and preterm birth has been shown to be one of the most consistent predictors of developmental status of children aged 24 and 36 months [30,31]. When we restricted these analyses to only those mother–child pairs with full term delivery, all associations were attenuated; however, the associations between early IL-8 and problem solving and fine motor skills retained nominal significance. A similar pattern was observed when adjusting for maternal pre-pregnancy BMI. In contrast to previous reports [28,29], we found that in this cohort prepregnancy BMI was not associated with developmental outcomes, and therefore inflammatory markers were not acting as mediators.

There were several limitations to these analyses. It has been argued that biomarkers drawn from maternal blood are only indirect indicators of fetal exposure and we do not have measures in the offspring. However, there is evidence that IL-8 can cross the placenta and blood–brain barrier [40], and that prenatal CRP and cytokine levels correlate with levels in the offspring in the weeks after birth [41,42]. Further, in this study we only had measures of two non-specific markers of inflammation, which may not be an accurate reflection of the multiple mechanisms involved in the inflammatory and immune response across pregnancy. However, both IL-8 and CRP have a strong body of literature implicating them in offspring development [6], justifying our investigation into these specific markers.

The ASQ relies on parental report, which carries the potential for bias; however, among similar questionnaires the ASQ has shown some of the strongest test characteristics and is recognized to provide an accurately identify children whose development is on schedule and children whose is not [43]. Parental report of their child’s development has been shown to be accurate and reliable, and therefore beneficial to, and appropriate for, assessment [44]. Nevertheless, we cannot exclude residual confounding relating to the parent’s mental wellbeing at the time of questionnaire completion, and for which we do not have information in this cohort. Although it is a screening rather than a diagnostic tool, the ASQ has been validated with high sensitivity and specificity for the identification of children under five with developmental delays [36]. Nevertheless, when considering the implications of these findings, it must be noted the majority of children scored well on all domains of the ASQ and these results may not be generalizable to more severe developmental delays. In keeping with previous evidence that it is unusual for children to show global developmental problems across all domains of the ASQ [39], only a very small number of VDAART children were in the lowest two categories for all domains, and there was no consistent evidence that these children were exposed to the highest levels of maternal inflammatory biomarkers. Due to the fact that a large number of children were missing a completed questionnaire at at least one of the three ages and because we did not see high correlation in ASQ domain scores across the first three years of life, we analyzed each age (one, two, three) independently. There were also some limitations in our potential confounding variables. Due to small numbers per category and to ensure model convergence, race was treated as a three-category variable (Black, White, Other) and we were unable to take ethnicity into account. Further, we used maternal marital status (together with maternal educational level) as a proxy for socioeconomic status, as we do not have more nuanced measures of socioeconomic status in this cohort. The parent cohort was designed to be enriched for asthma and allergies [45], which may themselves associate with child neurodevelopment and limit the generalizability of these findings to other populations [46]. We did not have information on time of day of blood collection, proinflammatory cytokines do exhibit a circadian rhythm [47], but we were unable to account for this in our analyses which may have influenced our findings. While CRP is not subject to diurnal variation [48], sleep deprivation and circadian misalignment, which can be common during pregnancy [49], can influence levels [50], but we are unable to quantify the impact of these factors on levels and therefore on our conclusions.

There were also several strengths to these studies. These analyses were conducted within a racially, ethnically and socially diverse and well-characterized US based cohort [45] with careful consideration of potential confounding and effect modifying factors. It is the first study to explore maternal measures of IL-8 and CRP and offspring development as assessed by the ASQ adding to the existing literature by focusing on a more broad and general measure of neurodevelopment than previous studies that have mainly focused on specific serious disorders. These results support the hypothesis that maternal levels of IL-8 in early pregnancy may associate with certain facets of neurodevelopment. Although the absolute decrease in development score per unit increase in biomarker level was small, the downstream effects could still be substantial and therefore studying these more general measures of development is of crucial importance. For example, it is known that the development of motor skills has an impact on early life beyond locomotion and body control as a child’s ability to move influences the means by which and the quality of a child’s interaction with their environment [5,43]. Accordingly, early neuro-motor dysfunction in children has been associated with working memory, processing speed and with academic, cognitive, and behavioral problems at later ages [43,51]. It has also been reported that children who walk earlier have had different affective relationships with their mothers than children who started walking later [43]. Communication and personal interaction skills in early life, are also of long-term importance as they form the basis of learning and social relationships for the rest of a child’s life [52]. Therefore, identifying factors that may influence them is of paramount importance.

## 4. Materials and Methods

### 4.1. Study Population

The Vitamin D Antenatal Asthma Reduction Trial (VDAART) was a designed as a randomized, double-blind, parallel-design trial to investigate whether prenatal vitamin D supplementation reduced the risk of asthma among offspring (ClinicalTrials.gov identifier: NCT00920621). Details on study rationale, design, methods, and results have been published previously [21,45]. In brief, pregnant non-smoking women aged 18–39 years of all races and socioeconomic status categories were recruited at a scheduled obstetrical prenatal visit from three sites across the United States between October 2009 and July 2011 (Boston Medical Center, Boston, MA, USA; Washington University at Saint Louis, St. Louis, MO, USA; Kaiser Permanente Southern California Region, San Diego, CA, USA). Women were recruited on the basis of a history of asthma or allergies in themselves or in the biological father, and who were an English or Spanish speaker and intended to participate for at least four years (up the to the third birthday of the child). At 10–18 weeks gestation, women were randomized to 4000 IU vitamin D daily plus a multivitamin containing 400 IU vitamin D or to a daily placebo and the same multivitamin. The women were followed monthly until the end of their pregnancy, and the children continue to be followed for a variety of health and wellness outcomes. The institutional review boards at each participating Clinical Center and the Data Coordinating Center at Brigham and Women’s Hospital approved protocols of the trial, with informed consent obtained from pregnant women at the enrollment visit covering primary and secondary analyses of data. The current analyses were restricted to mother–child pairs in whom the mother had blood samples in early and late pregnancy suitable for measurement of CRP and IL-8, and the offspring had a completed ASQ at ages one, two and/or three years.

### 4.2. Maternal Measures of IL-8 and CRP

Maternal levels of CRP and IL-8 were measured during the enrollment visit (10–18 weeks) and during the third trimester (32–38 weeks). Levels were quantified using the validated Meso Scale Discovery Platform (Rockville, MD, USA) assays at the Laboratory of Genital Tract Biology, Brigham and Women’s Hospital, under accreditation by the College of American Pathologists [31]. One hundred percent of the measured samples were above the assay lower limit of detection (LLD = 1.490 pg/mL for CRP and 0.0308 pg/mL for IL-8). Coefficients of variability were 5.04% for CRP and 6.98% for IL-8 [31].

### 4.3. The Ages and Stages Questionnaire

The ASQ 3rd Edition [22] was administered to the primary caregivers of the VDAART children at their age one-, two- and three-year follow-up visits. The questionnaire includes five developmental domains: gross motor skills, fine motor skills, problem solving ability, personal social skills, and communication. Each domain consists of six age-specific questions concerning a child’s ability to perform a task, with the parent responding “Yes” (10 points), “Sometimes” (5 points) or “Not yet” (0 points) which are summed to generate a domain score. Domain-specific scores are then compared to the expected mean score obtained from a reference distribution for that age group and categorized as: (i) “On Schedule for developing normally” (above the mean); (ii) “Requires Monitoring” (1–2 standard deviations below the mean); (iii) “Needs further evaluation” (>2 standard deviations below the mean).

### 4.4. Statistical Analysis

Pearson correlation was used to assess the relations between the two inflammatory markers with each other and across pregnancy. Similarly, Pearson correlation was used to assess the relations between ASQ scores across domains and ages.

Linear regression models adjusting for study site (Boston, St Louis, San Diego, CA, USA), treatment assignment (vitamin D or placebo), maternal race (Black, White, Other), maternal marital status (married or cohabiting, not living with father), maternal education (college graduate or higher, less than college education) and gestational age at delivery (weeks), were used to assess the association between log transformed CRP and IL-8, to account for skewness, with each domain score at ages one, two and three. Adjustment covariates were chosen based on the literature and a priori biological knowledge. As in previous studies of this cohort and others, maternal marital status and education were used as proxies for socioeconomic status. We further assessed the association between the change in biomarker level from enrollment to the third trimester with each domain score, using the delta of the biomarker level between the two blood draws. The Benjamini–Hochberg false discovery rate (FDR) was employed to account for multiple testing. To explore potential effect modification by offspring sex, breastfeeding status and treatment arm (vitamin D or placebo) in the biomarker-skills score association, we reran all models including a biomarker-interaction term. We ran additional sensitivity analyses excluding children born preterm and determined whether inflammatory biomarkers were mediating associations between maternal BMI ASQ scores using the R package ‘mediation’.

## 5. Conclusions

In summary, in this exploratory study we report that higher levels of IL-8 during pregnancy may influence problem solving and fine-motor skills in offspring, but not other facets of development. Furthermore, we report no associations with maternal CRP and offspring development. However, further work is needed to determine if other maternal inflammatory markers influence ASQ assessed child development, and we cannot rule out an association between prenatal inflammation and stress with offspring development. Given that many of the causes of inflammation, infection and prenatal stressors are modifiable, regardless, of these findings, future work should focus on the prevention and management of those factors, both social, emotional and clinical. By addressing these issues with the mothers both before and during pregnancy, we can act to prevent the potential lifelong downstream consequences in their children.

## Figures and Tables

**Figure 1 ijms-23-15249-f001:**
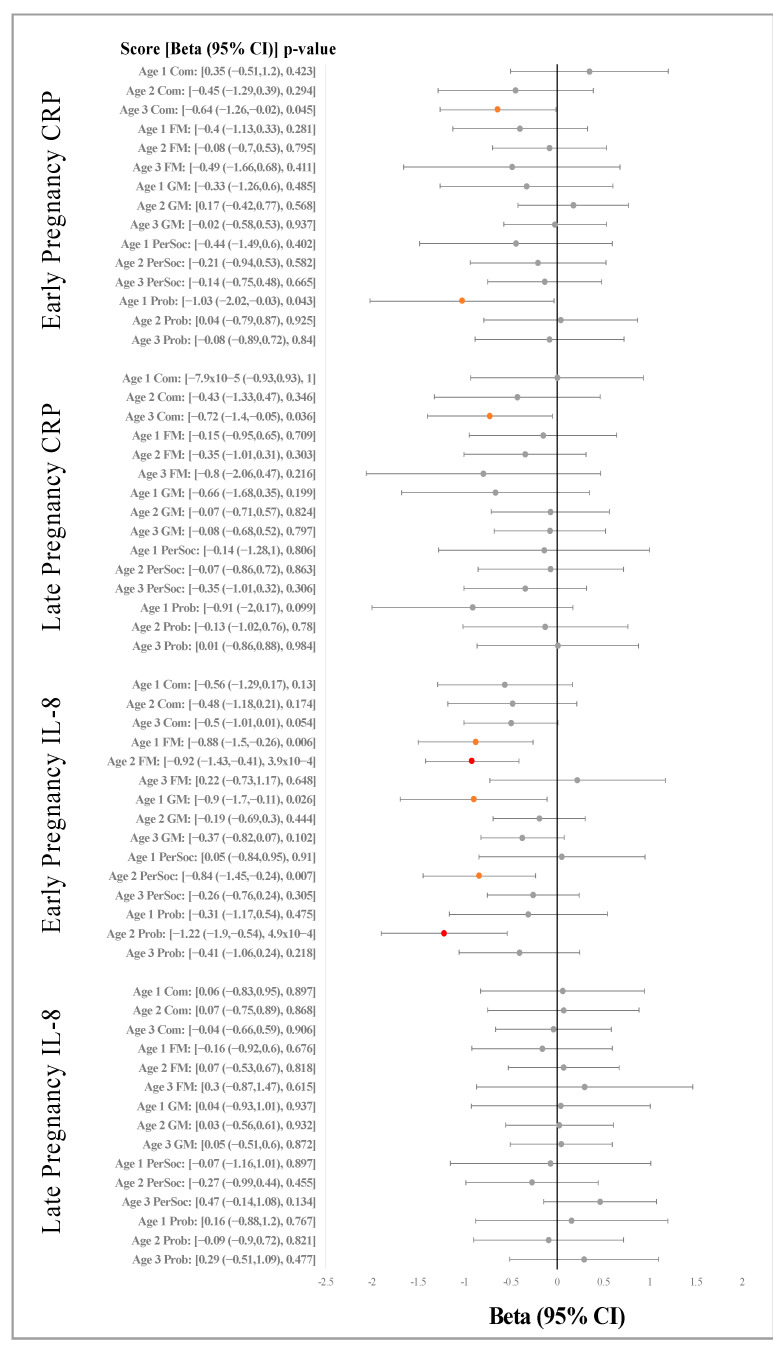
Associations between 1st and 3rd trimester levels of CRP and IL3 with skills scores in the five domains of the ASQ at ages 1, 2, and 3. ASQ score and biomarker levels were both treated as continuous variables. Analyses were adjusted for study site, treatment group, maternal marital status and educational level and gestational age at delivery. Grey: NS; Orange: Significant at *p* < 0.05; Red: Significant after Benjamini–Hochberg FDR correction; PerSoc: Personal-social score; Comm: Communication score; FM: Fine-Motor score; GM: Gross-Motor score; Prob: Problem-solving score.

**Figure 2 ijms-23-15249-f002:**
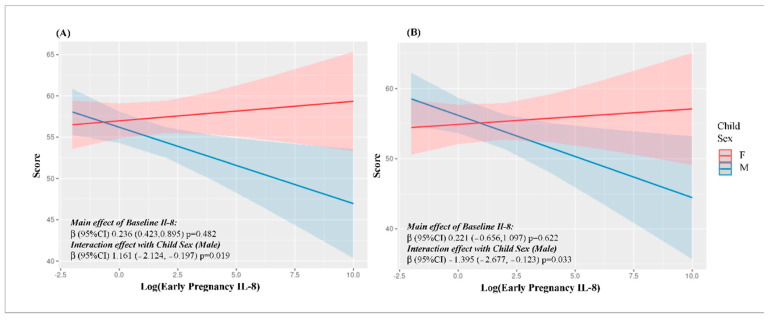
Interaction between log transformed IL8 levels in the first and third trimester and sex with (**A**) Personal-Social and (**B**) Problem-Solving skills at Age Three.

**Figure 3 ijms-23-15249-f003:**
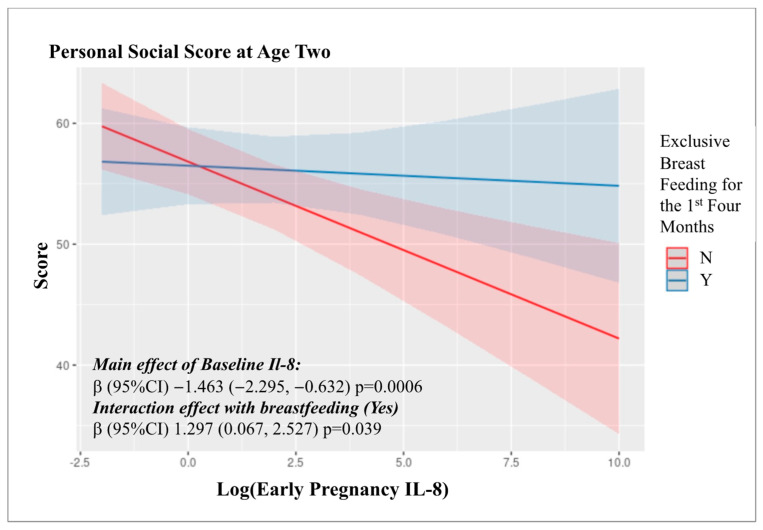
Interaction between log transformed IL8 levels in the first and third trimester and exclusive breast feeding up to four months with Personal-Social skills at Age Two.

**Figure 4 ijms-23-15249-f004:**
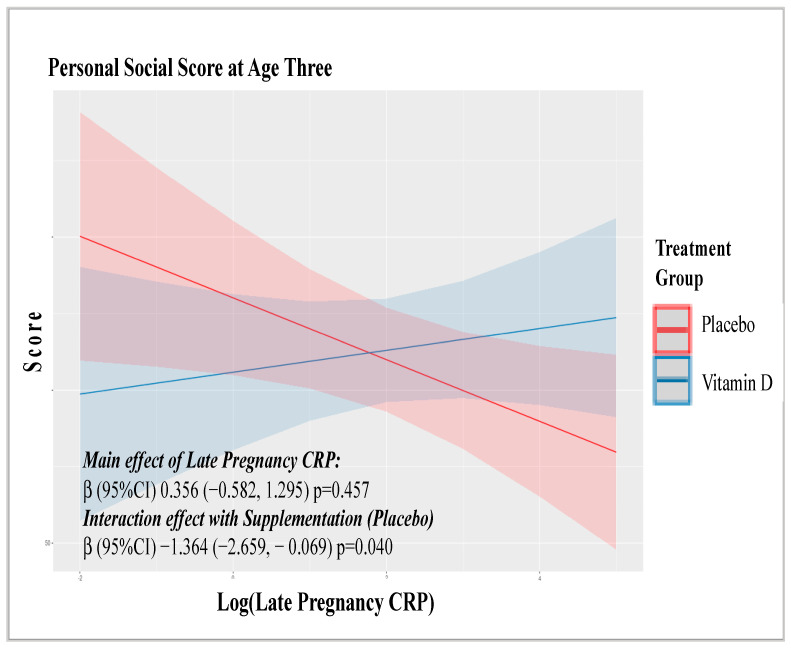
Interaction between log transformed IL8 levels in the first and third trimester and vitamin D supplementation with Personal-Social skills score at Age Three.

**Table 1 ijms-23-15249-t001:** Characteristics of 491 VDAART Mother–Child Pairs.

Variable			Mean/n	SD/%
Maternal Inflammatory Biomarkers	Early Pregnancy CRP	(mg/L)	2.09	1.12
	Late Pregnancy CRP	(mg/L)	2.02	1.04
	Early Pregnancy Trimester IL8	(pg/mL)	1.38	1.36
	Late Pregnancy Trimester IL8	(pg/mL)	1.25	1.10
Maternal Characteristics	Age at recruitment	(yrs)	27.7	5.4
	Race/Ethnicity	Black-Hispanic	21	4.3%
		Black-Non-Hispanic	187	38.1%
		Other-Hispanic	42	8.6%
		Other-Non-Hispanic	38	7.7%
		White-Hispanic	65	13.2%
		White-Non-Hispanic	138	28.1%
	Study Site	Boston	143	29.1%
		San Diego	169	34.4%
		St Louis	179	36.5%
	Education	Less than college grad	307	62.5%
		College grad or higher	184	37.5%
	Marital Status	Not living with Father	126	25.7%
		Married/Cohabiting	365	74.3%
	N. previous pregnancies at reruitment	0	173	35.2%
		1	125	25.5%
		2	91	18.5%
		3	50	10.2%
		4	27	5.5%
		>5	25	5.1%
	N. living children at recruitment	0	227	46.2%
		1	152	31.0%
		2	78	15.9%
		3	20	4.1%
		4	13	2.7%
		5	1	0.2%
	Treatment Arm	Placebo	254	51.7%
		Vitamin D	237	48.3%
Offspring Characteristics	Gestational age at delivery	(weeks)	39.2	1.59
	Gestational Age < 37 weeks	Yes	34	6.9%
	Sex	Female	242	49.3%
		Male	249	50.7%
	Race/Ethnicity ^a^	Black-Hispanic	40	8.1%
		Black-Non-Hispanic	193	39.3%
		White- Hispanic	62	12.6%
		White-Non-Hispanic	104	21.2%
		Other- Hispanic	54	11.0%
		Other-Non-Hispanic	38	7.7%
	Weight at birth	(grams)	3311.2	504.0
	Length at birth	(cm)	50.8	2.9
	Head circumference at birth	(cm)	34.1	1.9
	Exclusive breast feeding for first 4 months of life	Yes	157	32.0%
		No	300	61.1%
		Missing	34	6.9%

For continuous variables, mean and standard deviation (SD) are given. For categorical variables, number (n) and percent (%) are given. Early Pregnancy: 10–18 weeks; Late Pregnancy 32–38 weeks. ^a^ due to small numbers in each category for the multivariable regression models race was categorized as Black, White, Other.

## Data Availability

The data and materials necessary to reproduce the analyses presented here are not publicly accessible. The analytic code necessary to reproduce the analyses is available from the first author.

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
