# Peer review of "Maternal Inflammatory Biomarkers during Pregnancy and Early Life Neurodevelopment in Offspring: Results from the VDAART Study"

_ijms, 2022, doi:10.3390/ijms232315249_

Round 1
Reviewer 1 Report
This is a well written report describing an association between biomarkers of maternal stress (IL-8, CRP) measured at two timepoints in maternal blood during pregnancy. The results are clearly presented and there is extensive supporting data. The quality of the writing, presentation and discussion is excellent. The results are somewhat unsurprising and do not allow more than superficial inferences regarding underlying mechanisms. They are, however, consistent with previous work and therefore supportive of the general idea that maternal stress leads to adverse effects on the fetus. I have very few specific comments and questions, which mostly relate to possible additional confounders, as follows:
1. Proinflammatory cytokines exhibit a circadian rhythm. Is there information on what time of day blood samples were collected?
2. Are there data on maternal general health, infection history, social stress? Are there data on concurrent infections in pregnancy, antibiotic and analgesic use, fetal and placental birth weights and measurements?
3. How were women recruited to the Vitamin D / asthma study? Was this cohort typical of the general population?
4. Was there an interaction between CRP /IL8, offspring parameters, and Vitamin D administration?
5. Were these first pregnancies or randomly sourced? What was the prior experience of mothers with pregnancy? How many children were in the respective families?
Author Response
Reviewer One
This is a well written report describing an association between biomarkers of maternal stress (IL-8, CRP) measured at two timepoints in maternal blood during pregnancy. The results are clearly presented and there is extensive supporting data. The quality of the writing, presentation and discussion is excellent. The results are somewhat unsurprising and do not allow more than superficial inferences regarding underlying mechanisms. They are, however, consistent with previous work and therefore supportive of the general idea that maternal stress leads to adverse effects on the fetus. I have very few specific comments and questions, which mostly relate to possible additional confounders, as follows:
Thank you very much for taking the time to review this manuscript, we have addressed your comments below.
- Proinflammatory cytokines exhibit a circadian rhythm. Is there information on what time of day blood samples were collected?
This is an excellent point, but unfortunately this information was not collected in the VDAART study. We now discuss this important limitation in the ‘Discussion’ Section, with reference to how it may have influenced our findings for both IL-8 and CRP.
- Are there data on maternal general health, infection history, social stress? Are there data on concurrent infections in pregnancy, antibiotic and analgesic use, fetal and placental birth weights and measurements?
Thank you for raising these important questions. The parent VDAART clinical trial was focused on the effect prenatal supplementation of vitamin D on asthma in offspring, so unfortunately, we do not have as much information on the mothers as we would like. In particular, we did not ask questions regarding maternal stress or infection history (although information on antibiotic use was available, see below), nor do we have an overall measure of maternal health.
However, we have information on maternal asthma, atopic dermatitis/eczema and hay fever/allergic rhinitis and on associated medication use, as well as on several conditions of pregnancy.
Since this current manuscript under review was submitted a separate paper focused on preterm birth and CRP/IL-8 levels in this population has been published (PMID: 36241210), this paper additionally explored the association between maternal health during pregnancy and CRP/IL-8 levels. In the recently published paper we report that higher levels of CRP across pregnancy were significantly associated with lower maternal education, prior premature births, higher gravidity, uncontrolled asthma, unhealthy diet and risk of preterm birth. Higher levels of IL-8 were associated with a history of prior premature births, lower maternal education, an unhealthy maternal diet, and smoke exposure during pregnancy. We found no biomarker associations with perinatal antibiotic use, mode of delivery, vitamin D treatment assignment, or third trimester 25(OH)D levels.
Taken together these results suggest several medical conditions, a less healthy lifestyle and potentially a lower SES may associate with biomarker levels. As these results have now been reported in a different publication, we do not restate them in our results section, but we do now explicitly discuss these recently published findings in the discussion section citing the published paper.
However, we agree that considering fetal birth weight and measurements is a good idea. We did not have information on placental measurements, but we did have fetal measures.
We therefore sought to determine associations between CRP and IL-8 with fetal birth weight, head circumference and birth length. We found that birth weight was nominally associated with early pregnancy IL-8 (p=0.014) and CRP (p=0.0005), and that early pregnancy CRP was nominally associated with head circumference at birth (p=0.010).
However, birth weight, birth length and head circumference did not show any FDR significant associations with ASQ scores (there were only two nominally significant associations), therefore we decided that further investigating fetal measures were outside the scope of this manuscript and no additional results have been included in the revised manuscript.
We have now included fetal measures in Table 1 and Supplementary Table S1 to provide more context for the readers.
- How were women recruited to the Vitamin D / asthma study? Was this cohort typical of the general population?
We apologize for not providing enough information on the parent cohort. We have now expanded the section ‘4.1. Study Population’ to further explain how the women were reuited to VDAART and how they compare to the general population. We expand upon this in the dicussion, in partcular the fact that this cohort was recruited to be enriched for asthma and allergies and given that these two outcomes can themselves associate with neurodevelopment, how this may influence the generalisability of our findings. We also highlight that this cohort was diverse in terms of socioeceonmic status and race.
To specifically address how the sub population included in these analyses compare to the parent cohort, we have included a new supplementary table (Supplementary Table S1), comparing the parent and the subcohort. As can be seen from this table there were no significant differences between this subpopulation and the wider cohort. The only exception was in terms of gestational age which was significantly higher in the subpopulation. However, there was no difference in the percentage of children born preterm, and in general we do not believe this influences our overall findings or conclusions.
As noted in responses #2 and #5, we have also now expanded Table 1 and Supplementary Table S1 to provide information on number of previous pregnancies, number of living children, and weight, length and head cicrumderence at birth.
- Was there an interaction between CRP /IL8, offspring parameters, and Vitamin D administration?
In the recently published manuscript within this cohort (PMID: 36241210, response #2), we reported that first trimester maternal 25(OH)D levels were associated with and both first and third trimester CRP. However, there was no association with IL-8. We now report these findings in the current manuscript, citing this paper.
As suggested, we further sought to determine whether there was an interaction between CRP or IL-8 with vitamin D supplementation. We found only one significant interaction between late pregnancy CRP and age three personal social skills score (p-interaction=0.040), which suggests that the effects of maternal CRP may be more detrimental in those whose mothers received the placebo, but which were not robust to correction for multiple testing. We have now included these results in Supplementary Table S7 and Figure 4, and refer to them in the discussion
- Were these first pregnancies or randomly sourced? What was the prior experience of mothers with pregnancy? How many children were in the respective families?
We thank the reviewers for bringing up this important point, VDAART did not restrict recruitment to first pregnancies so the included mothers had previous pregnancies ranging in number from 0 to 13, and living children ranging from 0 to 5. This information has now been included in Table 1, and additionally in Supplementary Table S1 we demonstrate there was no difference in the number of pregnancies or children between the inflammatory biomarker subcohort and the parent VDAART study. We have previously shown (PMID: 36241210, response #2), that CRP levels were associated with higher gravidity and a higher number of living children, but IL-8 levels were not, therefore we do not report those findings here.
We did not run additional analyses to determine whether number of pregnancies of children was associated with ASQ scores. We identified one significant association between number of pregnancies and fine motor skills at age 2 (Beta: 0.514, 95% CI: 0.087, 0.942), p=0.018), but as this was not robust to correction for multiple testing, we did not investigate this finding further and do not include these results in the revised manuscript.
Reviewer 2 Report
The authors explored the potential influence of prenatal CRP and IL-8 on developmental outcomes as evaluated across the full breadth by the Ages and Stages questionnaire, which can be used as a screening tool to identify children in need of intervention.
The authors found that higher levels of IL-8 during pregnancy may influence problem solving and fine-motor skills in offspring.
The study is proper designed and analysed. The results presentation can be improved.
Table 2, Supplementary Table S5 and S6 is hard to read. Suggest use graph to present these data instead.
A few minor typos:
In the last sentence of Abstract "However, further reearch is required ..." should be "However, further research is required ..."
Page 3 line 94: "Several of the conditions ...." can be "Several conditions ....". More concise.
Author Response
Reviewer 2
The authors explored the potential influence of prenatal CRP and IL-8 on developmental outcomes as evaluated across the full breadth by the Ages and Stages questionnaire, which can be used as a screening tool to identify children in need of intervention.
The authors found that higher levels of IL-8 during pregnancy may influence problem solving and fine-motor skills in offspring.
The study is proper designed and analysed. The results presentation can be improved.
Thank you very much for this helpful review, we have made edits and alternations to address your concerns as detailed below
Table 2, Supplementary Table S5 and S6 is hard to read. Suggest use graph to present these data instead.
Thank you very much for this suggestion, we have now replaced the aforementioned Tables with Figures, which include both the effect estimates, 95%CI and p-values as numbers and also graphically in the form of a forest plot (Figure 1; Supplementary Figure S5 and Supplementary Figure S6).
We are happy to make any further edits to the figures/tables at the discretion of the editor.
A few minor typos:
In the last sentence of Abstract "However, further reearch is required ..." should be "However, further research is required ..."
Page 3 line 94: "Several of the conditions ...." can be "Several conditions ....". More concise.
Thank you very much for pointing these out, these have now been edited